https://doi.org/10.1038/s42003-019-0641-x — **OPEN**

# FRET-based cyclic GMP biosensors measure low cGMP concentrations in cardiomyocytes and neurons

Gaia Calamera[1,2], Dan Li[3], Andrea Hembre Ulsund[1,2], Jeong Joo Kim[4], Oliver C. Neely[3], Lise Román Moltzau[1,2], Marianne Bjørnerem[1,2], David Paterson [3], Choel Kim[4,5], Finn Olav Levy[1,2]* & Kjetil Wessel Andressen [1,2]

Several FRET (fluorescence resonance energy transfer)-based biosensors for intracellular detection of cyclic nucleotides have been designed in the past decade. However, few such biosensors are available for cGMP, and even fewer that detect low nanomolar cGMP concentrations. Our aim was to develop a FRET-based cGMP biosensor with high affinity for cGMP as a tool for intracellular signaling studies. We used the carboxyl-terminal cyclic nucleotide binding domain of *Plasmodium falciparum* cGMP-dependent protein kinase (PKG) flanked by different FRET pairs to generate two cGMP biosensors (Yellow *Pf*PKG and Red *Pf*PKG). Here, we report that these cGMP biosensors display high affinity for cGMP ($EC_{50}$ of $23 \pm 3$ nM) and detect cGMP produced through soluble guanylyl cyclase and guanylyl cyclase A in stellate ganglion neurons and guanylyl cyclase B in cardiomyocytes. These biosensors are therefore optimal tools for real-time measurements of low concentrations of cGMP in living cells.

[1] Department of Pharmacology, Institute of Clinical Medicine, University of Oslo and Oslo University Hospital, Oslo, Norway. [2] Center for Heart Failure Research, University of Oslo and Oslo University Hospital, Oslo, Norway. [3] Department of Physiology, Anatomy and Genetics, Oxford University, Oxford, UK. [4] Department of Pharmacology and Chemical Biology, Baylor College of Medicine, Houston, TX, USA. [5] Verna and Marrs McLean Department of Biochemistry and Molecular Biology, Baylor College of Medicine, Houston, TX, USA. *email: f.o.levy@medisin.uio.no

Cyclic guanosine 3′,5′-monophosphate (cGMP) is a second messenger involved in several intracellular signaling pathways in a variety of physiological functions and neurological and cardiovascular pathophysiology. Cyclic GMP is synthesized by NO-dependent activation of soluble guanylyl cyclase (sGC) and natriuretic peptide (NP)-dependent activation of particulate guanylyl cyclases; it activates protein kinase G (PKG) and cyclic nucleotide (CN)-gated ion channels and regulates phosphodiesterases (PDEs). The produced cGMP is degraded by PDEs. Cyclic GMP levels can be elevated by nitric oxide donors, sGC activators and stimulators, NPs and inhibition of specific PDEs, and several of these strategies protect the heart from pathological remodeling and ameliorate pulmonary arterial hypertension[1–5].

To investigate CN content, widely used methods to quantify CN production and degradation measures total intracellular content after cell lysis. However, this prevents measurement of the rapid dynamics and spatial intracellular organization of cGMP production and degradation. During the last decade, FRET (fluorescence/Förster resonance energy transfer) has been widely used to measure either protein–protein interactions or protein conformational changes through use of fluorescent biosensors. Such biosensors can be applied to detect CNs and consist of a CN-binding domain (BD), flanked by two fluorescent molecules. Upon CN-binding, the BD alters its protein conformation, thus changing the distance and increasing or decreasing FRET between the fluorophores[6,7]. Using such FRET-based biosensors, one can measure intracellular CN concentrations in real-time in single cells and overcome the limitations of traditional methods. Biosensors can either freely diffuse throughout the cell or be targeted to subcellular microdomains by tagging biosensors with specific proteins or short peptide sequences[7–9].

The dynamic measurement of intracellular cAMP concentrations has been successfully achieved by several intracellular FRET-based biosensors[7,8,10]. In contrast, dynamic measurements of intracellular cGMP have been challenging due to a paucity of FRET-based biosensors with high enough affinity for cGMP. Currently, the most frequently used biosensors for cGMP consist of a mammalian cGMP BD flanked by the fluorophores CFP and variants of YFP: Cygnet 2.1 (tandem CN BD and catalytic domains from PKG I with $EC_{50}$ ~1.7 μM)[11], cGi-500, cGi-3000 (only the tandem CN BD from PKG I with $EC_{50}$ ~0.5 μM and 3 μM, respectively)[12] and the cGES-DE5 (GAF (mammalian cGMP-binding PDEs, Anabaena adenylyl cyclases, and E. coli FhlA) domain of PDE5 with $EC_{50}$ ~1.5 μM)[13]. These biosensors with μM affinity have been used in cells where cGMP concentrations are relatively high, e.g. rat neonatal cardiomyocytes[14], smooth muscle cells[15] and mouse oocytes[16]. However, these biosensors have severe limitations in cells with low cGMP concentrations, which include adult cardiomyocytes and stellate ganglion (SG) neurons[17,18] where more sensitive biosensors are needed. For the cGES-DE5-based biosensor, exchange of fluorophores to T-sapphire and Dimer2 (Red cGES-DE5) increased the affinity for cGMP by ~35 fold[19], making it the only biosensor with high enough affinity to detect cGMP in cells with low cGMP, such as adult cardiomyocytes[17].

The protozoan parasite *Plasmodium falciparum* (*Pf*) is the cause of the most severe form of malaria[20]. Malaria transmission requires the gametogenesis phase within a mosquito. Although the signaling events that regulate the gametogenesis are not fully understood, cGMP is involved in promoting the gametocyte differentiation by activation of *Pf*PKG[21,22]. *Pf*PKG contains an N-terminal regulatory and a C-terminal catalytic domain displaying approximately 30–33% similarity with the mammalian PKGs. The most notable difference, compared with the mammalian PKGs, is the presence of four *Pf*PKG cGMP-BDs in the regulatory domain[22–24], compared to only two in mammalian PKGs. Moreover, the fourth cGMP BD (D-domain) displays high affinity (~40 nM) and selectivity for cGMP and serves as a gatekeeper domain for cGMP-dependent activation[25]. Recently, the apo- and cGMP-bound structures of the D-domain of *Pf*PKG were determined at high resolution[25]. Based on these structures, we constructed a FRET-based cGMP biosensor by flanking the *Pf*PKG D-domain with fluorescent protein pairs. We find that these biosensors display high affinity for cGMP and therefore could be utilized in cells with inherent low cGMP concentrations, such as cardiomyocytes and SG neurons.

## Results

**Development of high affinity FRET-based cGMP biosensors.** To design a high affinity cGMP biosensor, we analyzed the structure of the CN-binding domain D of *Pf*PKG (Fig. 1a, b) and found that the distance between the N- and C-termini (S403 and E542) was 35 Å in the absence of cGMP and increases to 56 Å upon cGMP binding (Fig. 1b). We therefore constructed a FRET-based biosensor by adding the fluorophores CFP and Venus (a YFP-variant) to the N- and C-terminus of the *Pf*PKG D-domain, respectively, and termed it Yellow *Pf*PKG (Fig. 1a). Next, we determined whether cGMP-binding altered FRET by expressing the biosensor in HEK293 cells and incubated homogenates with increasing concentrations of cGMP. Cyclic GMP decreased FRET (increased CFP/Venus ratio) in a concentration-dependent manner (Fig. 1c, d), suggesting that the N- and C-termini move further apart upon cGMP-binding. The affinity was in the nanomolar range ($EC_{50}$ of $22.5 \pm 2.8$ nM) and the maximal change in FRET at the highest cGMP concentration was $39.4 \pm 3.5\%$ (Fig. 1g, Table 1). To exclude whether basal cGMP from HEK293 cells interfered with our measurements, we constructed and purified a His-tagged Yellow *Pf*PKG biosensor and found a similar affinity for cGMP ($30 \pm 2$ nM; Supplementary Fig. 1) and maximal change in FRET ($42.3 \pm 0.4\%$). This indicates that the potential influence of endogenous cGMP from HEK293 cells on the determined cGMP affinity of our biosensor is negligible.

For the cGES-DE5 cGMP biosensor[13], replacing the CFP/YFP FRET pairs with T-sapphire/Dimer2 increased the affinity for cGMP by almost 40-fold[19]. To determine whether replacing the FRET pair of our Yellow *Pf*PKG biosensor could modify cGMP affinity, we constructed the Red *Pf*PKG biosensor, where CFP and Venus were replaced with T-sapphire and Dimer2 (Fig. 1a). The Red *Pf*PKG biosensor displayed similar affinity for cGMP ($EC_{50}$ $30.7 \pm 9.3$ nM) as the Yellow *Pf*PKG, but with a lower dynamic range ($27.9 \pm 2.7\%$ of the FRET change seen with the Yellow *Pf*PKG; Fig. 1e–g, Table 1). These results show that replacing the fluorophores does not necessarily change the binding affinity since the $EC_{50}$ was unchanged and secondly, that the T-sapphire and Dimer2 FRET pair produced a response with a lower dynamic range compared to the CFP/Venus FRET pair.

In the FRET-based cAMP biosensor based on EPAC1, replacing the acceptor (Venus) with a tandem $Cp^{173}$Venus-Venus increased the energy transfer efficiency and thus the dynamic range[26]. Therefore, to further improve the dynamic range of our Yellow *Pf*PKG biosensor, we replaced Venus with $Cp^{173}$Venus-Venus. We found a *reduction* in the dynamic range (>60% reduction; Table 1) and a possible reduction in cGMP affinity ($46.5 \pm 6.6$ nM; $p = 0.13$). Thus, simply adding a tandem $Cp^{173}$Venus-Venus to any biosensor does not enhance the magnitude of the FRET response.

**Real-time cGMP monitoring in single cells.** To determine if the Yellow and Red *Pf*PKG are functional in living cells, we first expressed the biosensors in HEK293 cells. As shown in Fig. 2a, b,

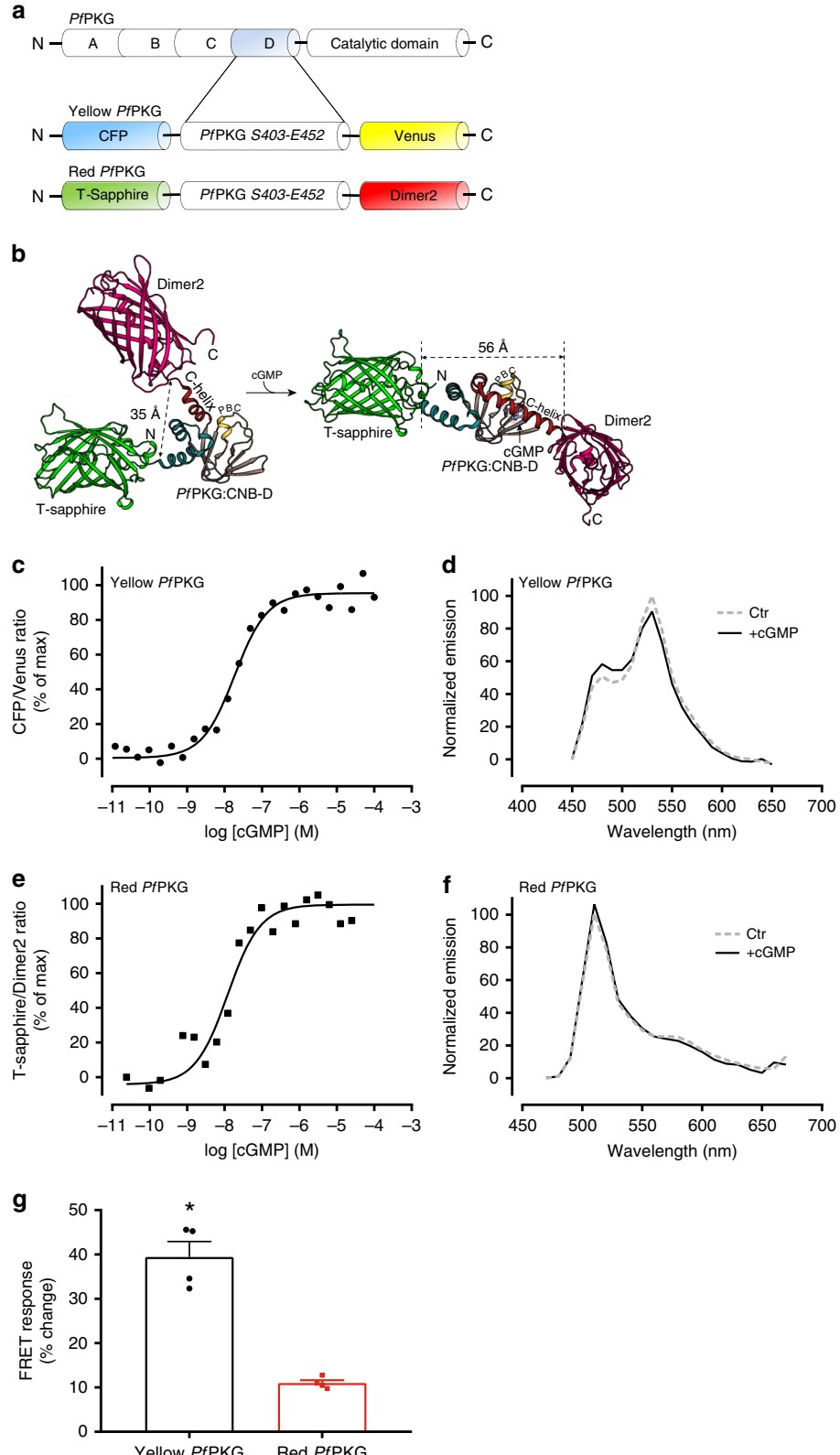

each fluorophore is expressed in HEK293 cells and diffusely, although not homogeneously, located throughout the cytoplasm. Thereafter, we monitored in real-time the cGMP produced in single cells through activation of sGC. Stimulating cells with the NO-donor SNAP (50 nM) increased FRET by 14.4 ± 1.1% for the Yellow PfPKG biosensor and 21.8 ± 4.5% for the Red PfPKG biosensor (Fig. 2c, d). Inhibiting PDE degradation of cGMP with

the non-selective PDE inhibitor IBMX (100 μM) further increased the FRET by 8.9 ± 6.9% for the Yellow PfPKG biosensor and 5.2 ± 2.4% for the Red PfPKG biosensor. Comparing the two biosensors, we found that the Yellow PfPKG displayed a significantly larger dynamic range in intact HEK293 cells compared to the Red PfPKG (41.5 ± 6.8% vs. 23.4 ± 2.6% maximal FRET change, respectively; p = 0.0002; Fig. 2e). We also measured total

**Fig. 1** Development of *Pf*PKG FRET biosensors for cGMP. **a** Schematic of the *Pf*PKG and the Yellow and Red *Pf*PKG biosensors. **b** The Red *Pf*PKG biosensor with the indicated conformational change upon cGMP binding. The D-domain of *Plasmodium falciparum* PKG (*Pf*PKG) is from S403 to E542 (PDB:4OFF and 4OFG; absence and presence of cGMP, respectively), T-sapphire (GFP) is from PDB:1GFL, Dimer2 (DsRed) is from PDB:1G7K. Similar structures would be expected from the Yellow *Pf*PKG. **c**, **e** Homogenates of HEK293 cells transfected with Yellow *Pf*PKG biosensor or Red *Pf*PKG biosensor (where indicated) were incubated with increasing concentrations of cGMP and FRET, displayed as $F_{CFP}/F_{Venus}$ (Yellow *Pf*PKG) and $F_{T\text{-}Sapphire}/F_{Dimer2}$ (Red *Pf*PKG) ratios, was measured as described in Materials and Methods and normalized to the minimum (absence of cGMP) and maximum (highest cGMP concentration). Shown are data from one representative of four independent experiments. **d**, **f** Emission spectra of the indicated biosensors from HEK293 homogenates incubated with (solid line) and without (dashed line) 100 μM cGMP and excited at 405nm. The emission spectra were normalized to maximum fluorescence in the absence of cGMP. **g** Maximal FRET response at the highest cGMP concentration. Data are mean ± SEM of four independent experiments. *$p = 0.0002$ vs. Red *Pf*PKG (Two-tailed Student's *t* test)

---

**Table 1 Affinity and dynamic range of *Pf*PKG biosensors for cGMP and cAMP**

| Sensor | cGMP | | cAMP | | Maximal FRET change (%) |
|---|---|---|---|---|---|
| | $p$EC$_{50}$ | EC$_{50}$ (nM) | $p$EC$_{50}$ | EC$_{50}$ (μM) | |
| Yellow *Pf*PKG | 7.66 ± 0.07 | 22 | 5.5 ± 0.2 | 4.6 | 39 ± 3* |
| Cp$^{173}$Venus-Venus *Pf*PKG | 7.34 ± 0.06** | 46 | 5.4 ± 0.3 | 5 | 13 ± 3 |
| Red *Pf*PKG | 7.58 ± 0.15 | 31 | 5.8 ± 0.2 | 2 | 10.9 ± 0.6 |
| Red *Pf*PKG Val536Gln | 7.75 ± 0.05 | 18 | 6.2 ± 0.2 | 0.7 | 15 ± 3 |
| Red *Pf*PKG Val536Glu | 7.59 ± 0.04 | 26 | 6.1 ± 0.1 | 0.8 | 16.0 ± 0.3 |
| Red *Pf*PKG Val536Asn | 7.65 ± 0.12 | 26 | 5.8 ± 0.1 | 2 | 18 ± 1 |
| Red *Pf*PKG Val536Asp | 7.76 ± 0.07 | 18 | 5.8 ± 0.1 | 1.5 | 12.1 ± 0.9 |

HEK293 cells transfected to express the indicated biosensors were homogenized and incubated with increasing concentrations of cGMP or cAMP and fluorescence measured as described in Materials and Methods. The maximal FRET change ($F_{CFP}/F_{Venus}$ or $F_{T\text{-}Sapphire}/F_{Dimer2}$) was determined at the highest cGMP concentrations. EC$_{50}$ shown in nM or μM were calculated from the average $p$EC$_{50}$ (-logEC$_{50}$). Results shown are mean ± SEM of 3-4 experiments
*$p < 0.001$ vs. all other biosensors
**$p = 0.13$ vs. Yellow *Pf*PKG (One-way ANOVA with Dunnett's post hoc test)

---

cellular cGMP and found that a higher concentration of SNAP (100 μM) was required to elicit a response and that CNP did not increase cGMP in HEK293 cells (Fig. 2f).

To determine whether our biosensors can detect low cGMP concentrations, we expressed the Yellow *Pf*PKG in cardiomyocytes where maximal cGMP concentrations are in the low nanomolar range[17]. To express Yellow *Pf*PKG in adult rat ventricular cardiomyocytes, we transduced cardiomyocytes with the biosensor using adenovirus (Fig. 3a). In contrast to the results obtained with HEK293 cells, stimulation of cardiomyocytes with a high concentration of SNAP (100 μM) did not alter FRET (Fig. 3b), despite increasing intracellular NO release (Supplementary Fig. 2). To exclude the possibility that constitutive sGC activation (or endogenous NO production) increases cGMP to concentrations that would saturate the biosensor, we incubated cells with an inhibitor of sGC (10 μM ODQ). ODQ did not modify FRET (Fig. 3c), suggesting that cGMP levels from activation of sGC are rather too low to be detected by the biosensor. In addition, measuring SNAP-stimulated total cGMP concentrations in cardiomyocytes only revealed a small pool of cGMP, apparently not accessible to our biosensor (Fig. 3d).

Next, we stimulated the particulate GCs in cardiomyocytes. While GC-A stimulation with BNP (300 nM) produced a modest FRET response (2.0 ± 0.4%, Fig. 3e, j), stimulation of GC-B receptors with CNP (300 nM) increased FRET by 32.8 ± 3.4% (Fig. 3f, j), consistent with other reports[17,27–29]. Adding IBMX (100 μM) further increased FRET by 5.4 ± 1.7%. From these results, we conclude that our biosensor detects cGMP in cardiomyocytes and secondly that GC-B-stimulation triggers a greater pool of cGMP compared to GC-A and sGC activation in cardiomyocytes, in line with our previous functional data[27,29,30].

Next, we compared our Yellow *Pf*PKG with the only other cGMP biosensor reported in adult cardiomyocytes, the Red cGES-DE5 biosensor[17,28]. CNP-stimulation of cardiomyocytes expressing Red cGES-DE5 increased FRET by 16.5 ± 3.1% (Fig. 3g, j). However, the Red cGES-DE5 displayed a significantly ($p =$

0.0001) lower dynamic range; 20.8 ± 2.1% (maximal FRET change; CNP + IBMX) compared to the Yellow *Pf*PKG (35.0 ± 2.1%, Fig. 3k). Since both the Yellow *Pf*PKG and the Red cGES-DE5 biosensors detect cGMP with similar affinities in vitro (Table 1 and Niino et al.[19]), we determined their response to increasing concentrations of CNP in intact cardiomyocytes (Fig. 3h, i) and found a relatively similar potency of CNP with the two biosensors ($p$EC$_{50}$ 7.4 ± 0.2 for the Yellow *Pf*PKG and 7.1 ± 0.1 for the Red cGES-DE5; $p = 0.37$). This indicates that the Yellow *Pf*PKG biosensor has similar or slightly higher sensitivity for cGMP compared to established biosensors and yields a larger dynamic range.

Previously, we used the cGi-500 cGMP biosensor in SG neurons and observed that BNP stimulation of GC-A increased cGMP and reduced noradrenaline release[18]. However, cGi-500 only showed a modest signal after GC-A stimulation, suggesting low levels of cGMP. Therefore, we tested our Yellow *Pf*PKG biosensor to better visualize the low cGMP produced by BNP stimulation. We performed real-time measurements of cGMP by expressing either the cGi-500 or the Yellow *Pf*PKG biosensor in SG neurons. Adding increasing concentrations of BNP (10, 100, and 250 nM) increased FRET (Fig. 4a–c). The highest concentration of BNP (250 nM) increased FRET more for the Yellow *Pf*PKG than for the cGi-500 biosensor ($p = 0.01$). Next, we activated sGC by applying the sGC activator BAY 41-2272 and found that both biosensors gave an increase in FRET (Fig. 4d–f). To determine the dynamic range of both biosensors in SG neurons, we saturated cGMP levels by stimulating with both the NO donor Sin-1 (20 μM) and IBMX (100 μM). The maximal FRET change observed for the cGi-500 was significantly higher ($p < 0.0001$) than the Yellow *Pf*PKG biosensor (Fig. 4g).

**PfPKG biosensors display 100-fold selectivity**. Next, we determined the cGMP selectivity by incubating the *Pf*PKG biosensors with increasing concentrations of cAMP and measuring FRET (Fig. 5a, b). We found that cAMP increased FRET at ~100-fold

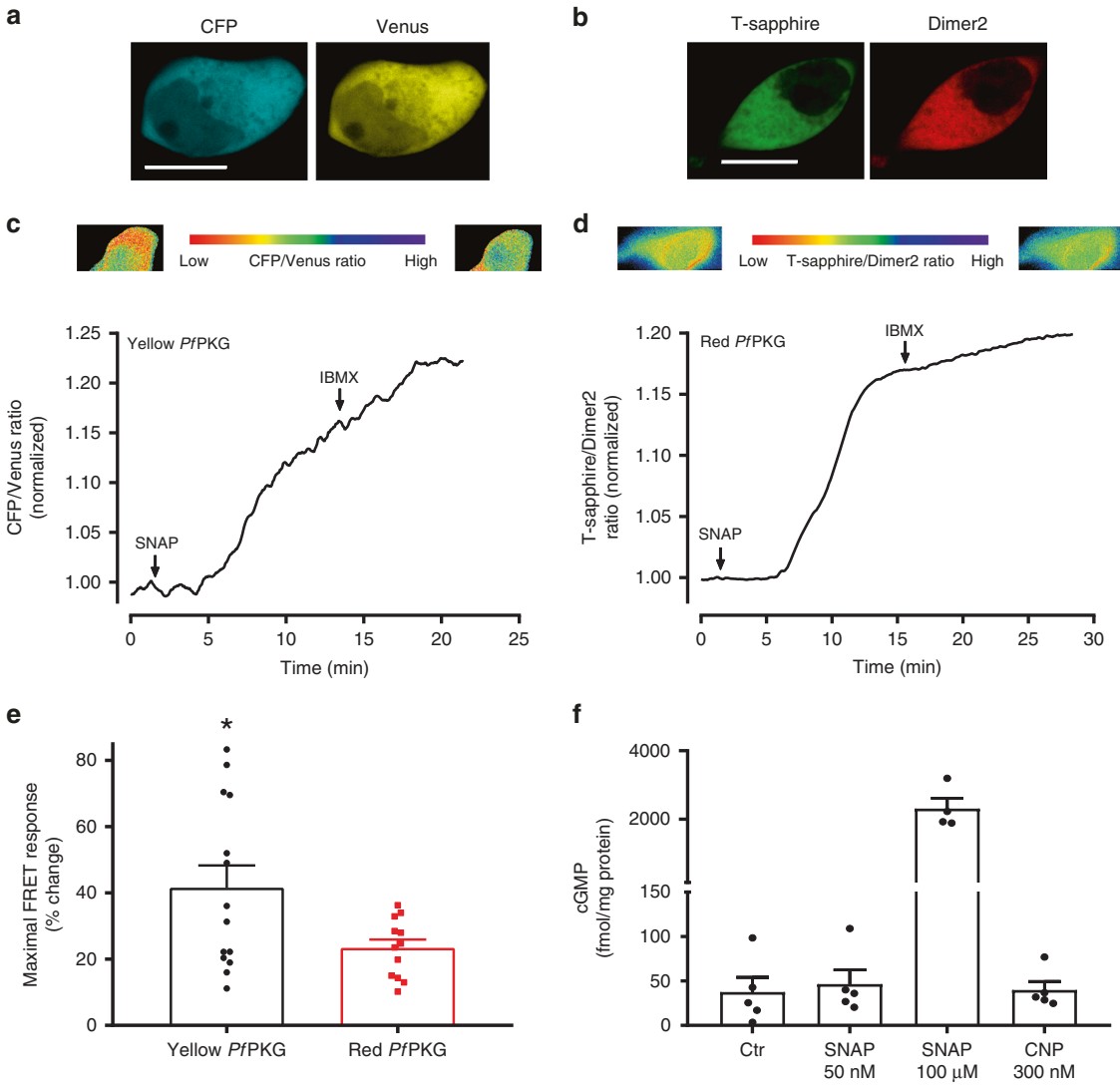

**Fig. 2** *Pf*PKG biosensors detect sGC-stimulated cGMP increase in HEK293 cells. **a** Confocal images of living HEK293 cells expressing the Yellow *Pf*PKG biosensor. Scale bar: 10 μm. **b** Confocal images of living HEK293 cells expressing the Red *Pf*PKG biosensor. Scale bar: 10 μm. **c**, **d** Recording of FRET ratio between CFP and Venus (c) or T-sapphire and Dimer2 (d) in single HEK293 cells expressing the indicated biosensor and stimulated with the NO donor SNAP (50 nM) and the PDE inhibitor IBMX (100 μM), where indicated. FRET was normalized to that prior to stimulation. Shown are traces representative of four (Yellow *Pf*PKG) and five (Red *Pf*PKG) individual cells. Pseudocolour images show CFP/Venus and Tsapphire/Dimer2 ratios at the start and end of experiment. **e** Maximal FRET response at the saturating stimulation (SNAP + IBMX). Data are mean ± SEM from 14 (Yellow *Pf*PKG) and 12 (Red *Pf*PKG) cells. *$p = 0.027$ vs. Red *Pf*PKG (Two-tailed Student's $t$ test). **f** cGMP levels in HEK293 cells in the absence (Ctr) or presence of 50 nM or 100 μM SNAP (10 min stimulation). Data are mean ± SEM from 4 to 5 individual experiments

higher concentrations compared to the cGMP response (cAMP EC$_{50}$ of 4.6 ± 2.1 μM and 1.9 ± 0.6 μM for the Yellow and Red *Pf*PKG biosensors, respectively). To determine if cAMP levels in HEK293 cells were detected by the Yellow *Pf*PKG, we stimulated endogenous β-ARs in HEK293 cells[31] with isoprenaline (10 μM) and found a 3.7 ± 0.5% increase in FRET (Fig. 5c). In separate experiments, direct activation of adenylyl cyclases by forskolin (25 μM) increased FRET by 13.8 ± 2.8% (Fig. 5d). However, in both cases, further stimulation with SNAP (50 nM) gave an additional increase in FRET (11.6 ± 2.8% after isoprenaline and 29.5 ± 5.1% after forskolin), excluding the possibility that the biosensor was fully saturated (Fig. 5c, d). Together, these results suggest that the biosensor is only sensitive to high cAMP concentrations. To determine whether this was also the case in cardiomyocytes, we incubated these with isoprenaline (10 nM) and obtained a 10.9 ± 2.6% transient change in FRET. Additional stimulation with CNP (300 nM) gave a further increase of 19.4 ±

3.2% and subsequent stimulation with IBMX (100 μM) gave an additional 2.6 ± 0.7% increase in FRET (Fig. 5e, g). The Red cGES-DE5 biosensor, on the other hand, did not detect a similar cAMP increase (Fig. 5f, g).

Since this could pose a problem for measuring cGMP within the cells where cellular cAMP levels are 10–100 fold higher than the cGMP levels, we attempted to reduce the affinity towards cAMP without altering the affinity for cGMP. Previously, the Zagotta group demonstrated that mutation of an isoleucine residue to aspartic acid (Ile636Asp) in an HCN ion channel of sea urchin sperm (SpIH channel), resulted in a cGMP selective and sensitive SpIH channel[32]. Aligning the *Pf*PKG D-domain with the CN-BD of the mutant SpIH channel (Ile636Asp) showed that Val536 of *Pf*PKG is analogous to Ile636. Thus, we hypothesized that replacing Val536 with polar or charged amino acids (glutamine, glutamic acid, asparagine or aspartic acid) could improve cGMP selectivity. We therefore mutated Val536 to Gln,

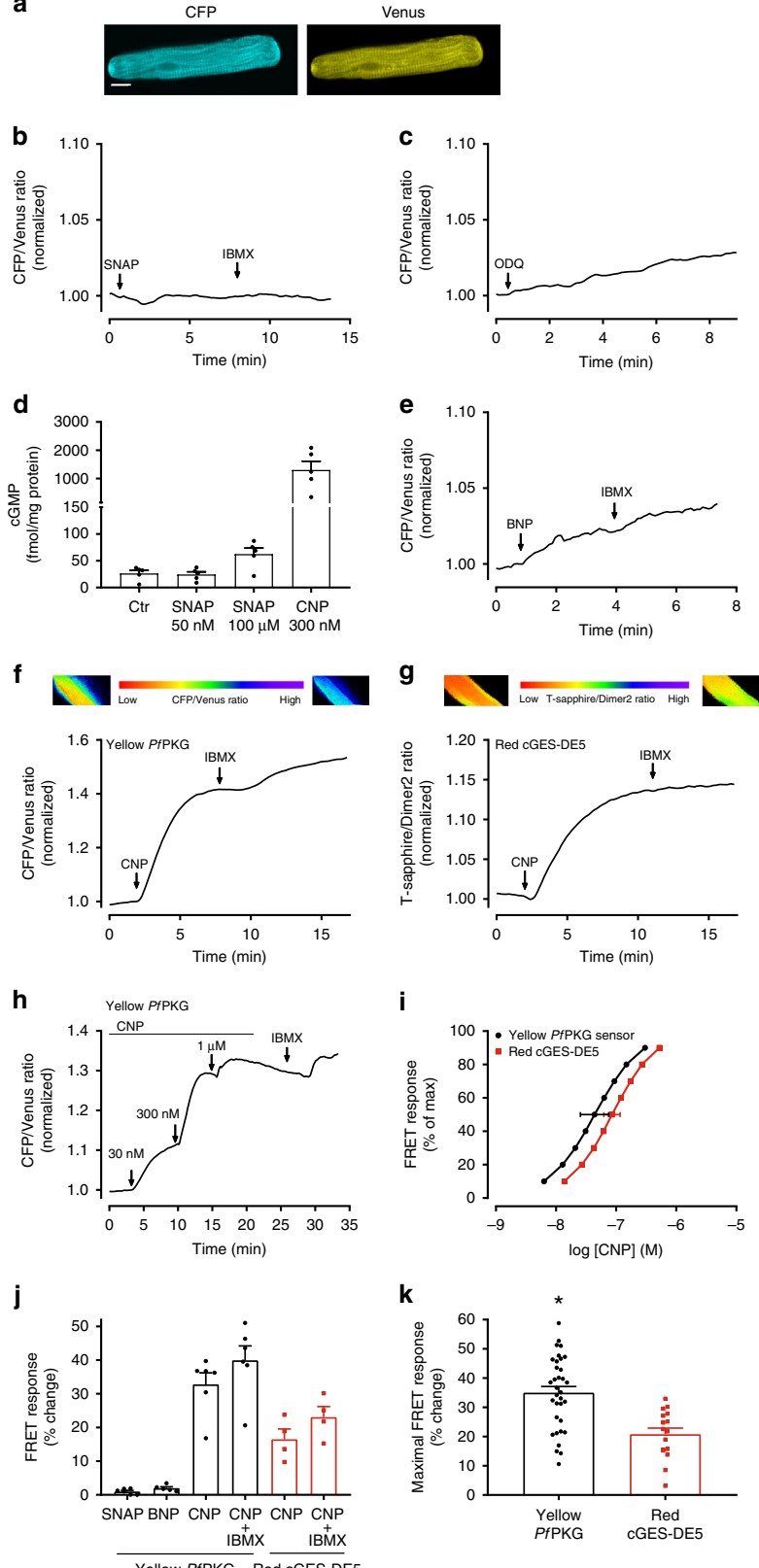

Glu, Asn, and Asp. Although the affinity for cGMP was retained, the selectivity did not improve, as both CN affinities were essentially unaltered (Table 1). This suggests that other molecular determinants within the *Pf*PKG D-domain are responsible for cGMP selectivity compared to the SpIH channel.

## Discussion

In this paper, we developed previously unreported FRET-based biosensors that can monitor cGMP dynamics in real time in intact cells with low intrinsic cGMP levels. The Yellow *Pf*PKG biosensor displays nanomolar affinity for cGMP and a large

**Fig. 3** Yellow *Pf*PKG biosensor monitors cGMP from GC-A and GC-B in cardiomyocytes. **a** Confocal images of a living cardiomyocyte expressing the Yellow *Pf*PKG biosensor. Scale bar: 10 μm. **b**, **c**, **e**, **f** Recording of FRET ratio (CFP/Venus) in single cardiomyocytes expressing the Yellow *Pf*PKG biosensor and stimulated with the NO donor SNAP (100 μM), the PDE inhibitor IBMX (100 μM), the sGC inhibitor ODQ (10 μM), BNP (300 nM) or CNP (300 nM), where indicated. FRET signal was normalized to that at the time of stimulation. Shown are traces representative of six cells from four animals (SNAP), eight cells from four animals (ODQ), five cells from four animals (BNP) or six cells from six animals (CNP). **d** Total cGMP levels in isolated ventricular cardiomyocytes in the absence (Ctr) or presence of 50 nM SNAP, 100 μM SNAP or 300 nM CNP (10 min stimulation). Data are mean ± SEM from five individual experiments. **g** Recording of FRET ratio (T-sapphire/Dimer2) in single cardiomyocytes expressing the Red cGES-DE5 biosensor and stimulated with CNP (300 nM) and IBMX (100 μM) (where indicated). FRET was normalized to that prior to stimulation. Shown are traces representative of four cells from four animals. **h** Recording of FRET ratio (CFP/Venus) in single cardiomyocytes expressing the Yellow *Pf*PKG biosensor stimulated with increasing concentrations of CNP. Shown are traces representative of nine individual cells from five animals normalized to that prior to stimulation. **i** Concentration–response curves generated by stimulation with increasing concentrations of CNP in single cardiomyocytes expressing the indicated biosensor. Data are plotted as centiles and for the EC$_{50}$ the mean ± SEM is indicated (nine cells from five animals for the Yellow *Pf*PKG, six cells from six animals for the Red cGES-DE5). $p = 0.37$ (unpaired t-test). **j** Quantification of FRET responses shown in b, c, and e–g. $p = 0.006$ for CNP-stimulation for *Pf*PKG vs. Red cGES-DE5 (one-way ANOVA with Sidak's multiple comparisons test). **k** Maximal FRET response (CNP+IBMX) from cardiomyocytes expressing the indicated biosensor. Data are mean ± SEM from 34 cells from 27 animals (Yellow *Pf*PKG) and 16 cells from 10 animals (Red cGES-DE5). *$p = 0.0001$ vs. Red cGES-DE5 (Two-tailed Student`s t test)

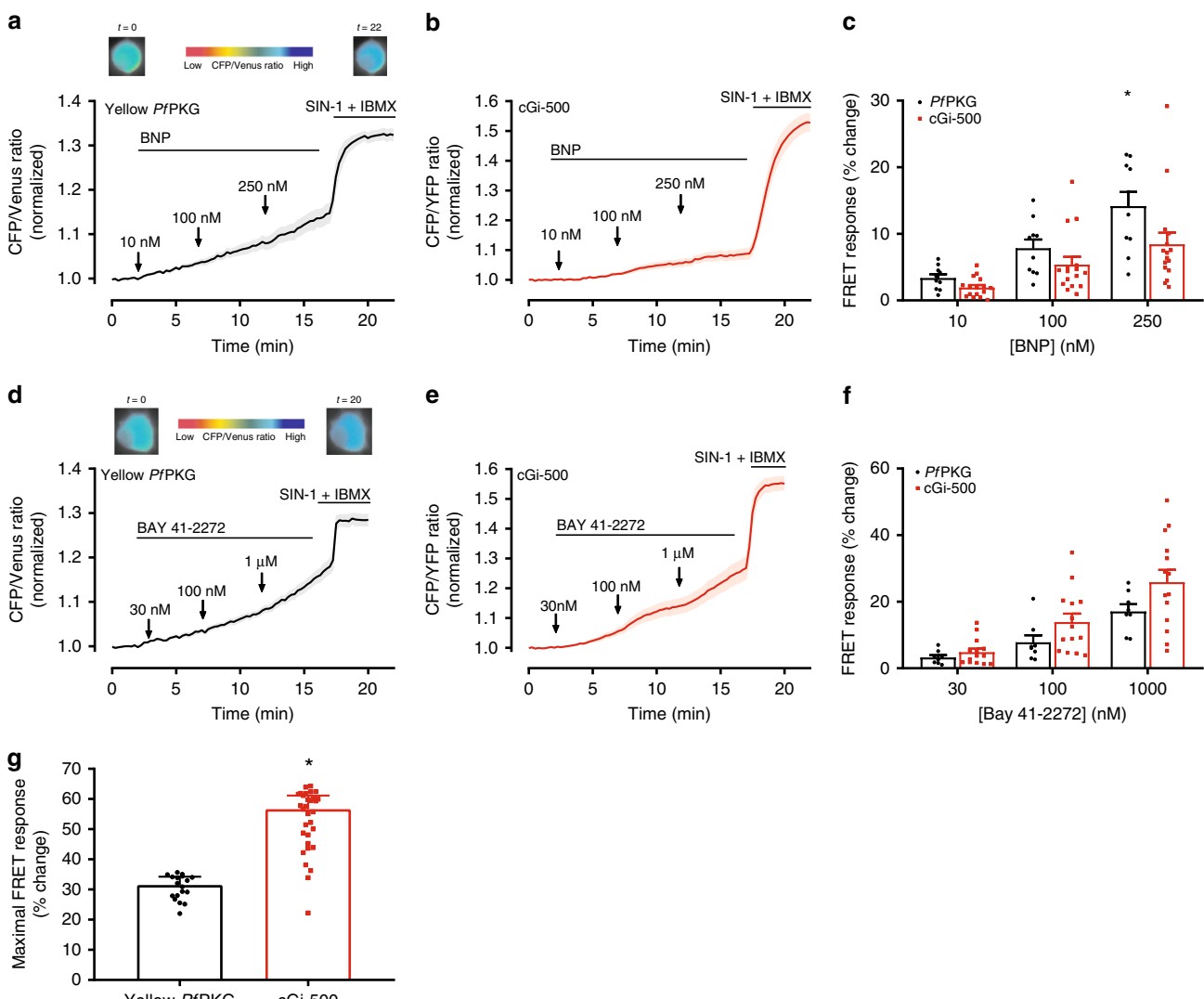

**Fig. 4** Yellow *Pf*PKG detects low cGMP from GC-A stimulation in stellate ganglion neurons. **a**, **b**, **d**, **e** Recording of FRET ratio (CFP/Venus for *Pf*PKG, CFP/YFP for cGi-500) in single stellate ganglion neurons expressing the indicated biosensors and stimulated with either the GC-A agonist BNP or the sGC activator BAY 41-2272 at increasing concentrations. The responses were then maximized by stimulating with SIN-1 (20 μM) and IBMX (100 μM). FRET signal was normalized to that prior to stimulation. Traces are mean ± SEM from 10 (*Pf*PKG) and 16 cells (cGi-500) in (**a**) and (**b**) and eight (*Pf*PKG) and 14 cells (cGi-500) in (**d**) and (**e**). **c**, **f** Quantification of FRET responses in (**a**), (**b**), (**d**), and (**e**). Data are mean ± SEM. *$p = 0.012$ cGi-500 vs. *Pf*PKG for BNP (250 nM) (Two-way ANOVA assuming sphericity with Sidak's multiple comparisons test). **g** Maximal FRET response at the saturating stimulation (SIN-1 20 μM+IBMX 100 μM). Data are median with interquartile range from 18 (*Pf*PKG) and 31 (cGi-500) cells. *$p < 0.0001$ vs. *Pf*PKG biosensor (Two-tailed Mann–Whitney test)

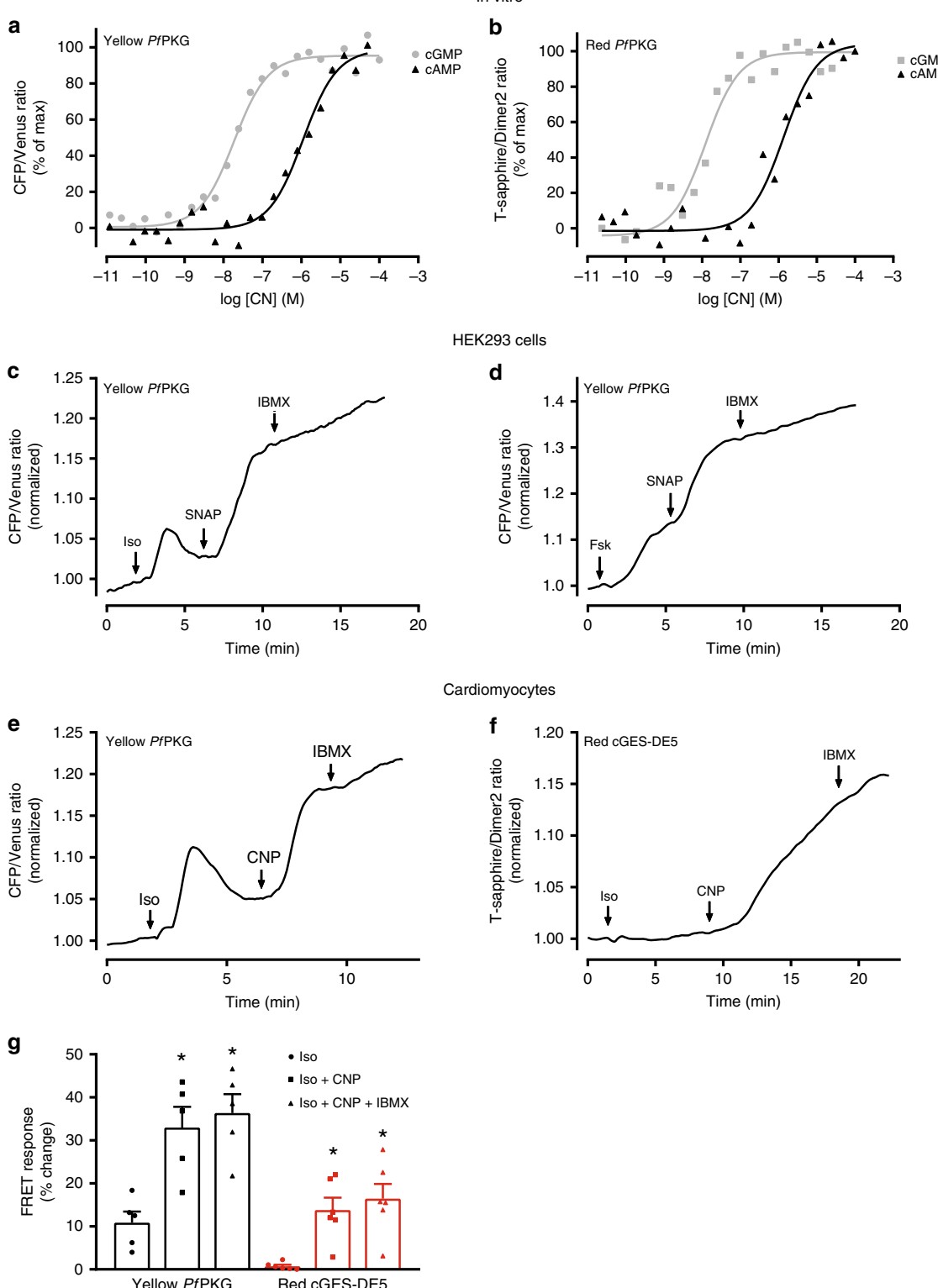

dynamic range. In addition, we detected GC-A- and GC-B-stimulated cGMP in cardiomyocytes and SG neurons, both examples of cells that have low cGMP concentrations.

The two biosensors described herein (Yellow and Red *Pf*PKG) differed only in terms of their efficacy, where the Red *Pf*PKG displayed a smaller dynamic range (Figs. 1g, 2e and Table 1). A similar reduction in dynamic range was also reported for the red vs. yellow versions of the cGES-DE5 biosensor[19]. This might be due to a different relative change of energy transfer upon cGMP binding in CFP/Venus compared to the T-Sapphire/Dimer2 fluorescent pairs. However, despite a lower dynamic range, the Red *Pf*PKG could still offer other advantages: the emission spectrum of T-sapphire/Dimer2 shows a reduced overlap compared to CFP/Venus, and thus a reduced bleed-through of donor in the acceptor channel[19], that increases the signal-to-noise ratio; cells have less autofluorescence at higher wavelengths[33] (emission

**Fig. 5** *Pf*PKG biosensors display 100-fold selectivity of cGMP over cAMP. **a**, **b** Homogenate of HEK293 cells transfected with Yellow *Pf*PKG biosensor (**a**) or Red *Pf*PKG biosensor (**b**), incubated with IBMX (100 μM) and increasing concentrations of cAMP (black). Corresponding data for cGMP (gray) from Figs. 1c, 1e are included for direct comparison. FRET was measured as described in Materials and Methods and normalized to the minimum (absence of CN) and maximum (highest CN concentration). Data shown are representative of four independent experiments. **c**, **d** Recording of FRET ratio (CFP/Venus) in single HEK293 cells expressing the Yellow *Pf*PKG biosensor and stimulated with isoprenaline (Iso; 10 μM) or the AC activator forskolin (Fsk; 25 μM) followed by SNAP (50 nM) and IBMX (100 μM). FRET signal was normalized to that prior to first stimulation. Data shown are representative of five (Iso-SNAP-IBMX) and five (Fsk-SNAP-IBMX) independent experiments. **e**, **f** Recording of FRET ratio in single cardiomyocytes expressing the indicated biosensor and stimulated with Iso (10 nM), followed by CNP (300 nM) and IBMX (100 μM). FRET signal was normalized to that prior to first stimulation. Data are representative of five cells from four animals (**e**) and six cells from four animals (**f**). **g** Quantification of data in e-f. Data are mean ± SEM. *$p < 0.05$ vs. "Iso" (two-way ANOVA with Tukey post hoc test)

537/623 nm vs. 470/530 nm for CFP/YFP), giving lower back-ground noise; and longer emission wavelengths are favorable for deep-tissue imagining[34,35], thus improving application of FRET imaging in tissues.

Stimulation of single HEK293 cells with 50 nM SNAP increased cGMP when using our biosensors (Fig. 2c, d). However, in experiments determining cGMP in whole cell lysates, this concentration of SNAP yielded barely detectable increases of cGMP (Fig. 2f), and higher concentration of SNAP was required to reveal a large cGMP increase in our global cGMP assay. This suggests that the *Pf*PKG biosensors offer more sensitive tools to detect low concentrations of cGMP. Therefore, in cells where sGC-, GC-A- or GC-B-stimulated cGMP levels are low, such as cardiomyocytes[17], a biosensor with high cGMP affinity could be suitable. In fact, the *Pf*PKG biosensor detected a large increase in cGMP after activation of GC-B by CNP and a small increase in cGMP after GC-A stimulation (Fig. 3j). Similar observations were also made in mouse cardiomyocytes using the Red cGES-DE5 biosensor[17,28]. Comparing the Yellow *Pf*PKG and Red cGES-DE5 biosensor, we observed similar sensitivity to CNP stimulation, consistent with similar affinities for cGMP (Table 1 and Niino et al.[19]), but the Yellow *Pf*PKG biosensor displayed a larger dynamic range (Fig. 3k). The $pEC_{50}$ for CNP is 7.4 ± 0.2, corresponding to an $EC_{50}$ value of about 41 nM for the Yellow *Pf*PKG biosensor (Fig. 3i) and we have previously reported that CNP induced a negative inotropic response and positive lusitropic response in rat muscle strips with a similar $EC_{50}$[36]. Also, CNP-induced modulation of β-AR-mediated positive inotropic response through PDE3 inhibition displayed a similar $EC_{50}$[29]. This suggests that the physiological effects of CNP are well within the cGMP concentration range detected by the Yellow *Pf*PKG biosensor.

It is well known that NO and sGC are present in cardiomyocytes and increase cGMP[37]. However, the detection of NO-stimulated cGMP by FRET-based biosensors has been a challenge to others[17] and despite the high cGMP affinity of the *Pf*PKG biosensors, they could not detect any cGMP even after 100 μM SNAP (Fig. 3b). However, in our measurements of total cellular cGMP content, this concentration of SNAP resulted in a small increase of cGMP in cardiomyocytes (Fig. 3d). We therefore speculate that the cGMP produced by sGC is restricted to compartments where our biosensor cannot get access. Interestingly, the confocal images show that the *Pf*PKG biosensor is not expressed evenly throughout the cardiomyocytes (Fig. 3a), suggesting that their expression or movement is somewhat limited. A similar non-uniform distribution pattern is also clearly seen with other non-targeted biosensors[7,17,38–40]. Perhaps such untargeted biosensors do not have access to certain intracellular structures or organelles where tight spatial signaling, such as that from sGC, could be confined.

Cardiac function can be modulated by the autonomic nervous system through a complex neuronal network. The sympathetic SG neurons innervate the sino-atrial node and myocardium and

release noradrenaline that induces chronotropic and inotropic responses in the heart[41]. BNP released from the heart is a compensatory mechanism in conditions such as heart failure and hypertension[42]. We have previously shown that BNP decreases noradrenaline release, but we detected only very limited increases in cGMP by using established FRET-based biosensors[18]. Thus, with the *Pf*PKG biosensor we revisited the cGMP dynamics upon stimulating SG neurons with BNP. The *Pf*PKG biosensor successfully detected the BNP-stimulated cGMP increase with a larger change in FRET compared to the cGi-500 (Fig. 4a–c). This is consistent with the >20-fold higher affinity of *Pf*PKG for cGMP. However, the dynamic range of the cGi-500 was larger than the *Pf*PKG biosensor (Fig. 4g). Thus, when comparing the two biosensors as per cent of maximal response, the difference between the biosensors in detecting the BNP effect was more accentuated (Supplementary Fig. 3). The reduced dynamic range is less likely due to the fluorescent pair, as they both share the same donor (CFP) and the *Pf*PKG contains a different acceptor (Venus) which is ~30 times as bright as the YFP[43] in cGi-500. However, the fluorophores of *Pf*PKG are attached to S403 and E542 and the distance between these is only increased by 21Å upon cGMP binding (Fig. 1b) which could account for a smaller change in FRET.

Considering the structural similarity between cGMP and cAMP and that increasing cGMP might inhibit PDE3-mediated degradation of cAMP[44], it is important to determine the selectivity of our biosensor. In vitro measurements show a selectivity of 100-fold for cGMP over cAMP and this is in accordance with the previous $EC_{50}$ values seen in the isolated D-domain using fluorescence polarization assays[25]. In live single cell measurements, stimulation of cAMP production resulted in a small response of the *Pf*PKG biosensor in both HEK293 cells and cardiomyocytes (Fig. 5). However, the biosensor was not saturated, as we could obtain a further change in FRET after stimulation of sGC (HEK293 cells) or GC-B (cardiomyocytes), suggesting that high concentrations of cAMP may elicit a modest response. Given that the $K_m$ of PDE3 for cGMP and cAMP is 0.1–0.8 μM[45], cGMP would start to inhibit PDE3 at concentrations where our biosensor is saturated. Additionally, the $EC_{50}$ of CNP was similar between the *Pf*PKG and the cGMP-selective Red cGES-DE5 biosensor. Taken together, the potential increases in cAMP levels by cGMP-mediated PDE3 inhibition will not be detected by the *Pf*PKG biosensor. However, in experiments where both cGMP and cAMP levels are simultaneously elevated[46], the *Pf*PKG biosensor alone cannot discriminate between low levels of cGMP and high levels of cAMP. To control for this, the Yellow *Pf*PKG biosensor would be suitable to use in combination with a cAMP biosensor, such as the Pink Flamindo[47], a single fluorophore biosensor with excitation (567 nm) and emission (590 nm) that do not interfere with the Yellow *Pf*PKG emission spectrum.

To conclude, we have generated two previously unreported biosensors for cGMP with high affinity and one of these display a

large dynamic range. The biosensors contain the same BD for cGMP but with different fluorophores and could therefore be applied for multiparameter fluorescence imaging which requires the simultaneous combination of different FRET biosensors.

## Methods

**Materials.** S-Nitroso-N-acetyl-DL-penicillamine (SNAP) and 1H-[1,2,4]Oxadia-zolo[4,3-a]quinoxalin-1-one (ODQ) were from Tocris Bioscience (Bristol, UK); 4-Amino-5-methylamino- 2′,7′-difluorofluorescein diacetate (DAF-FM-DA) was from abcam (Cambridge, UK); human C-Type Natriuretic Peptide (CNP) and rat Brain Natriuretic Peptide (BNP) were from GenScript Corp. (Piscataway, NJ,USA); All other chemicals and media were from Sigma-Aldrich (Sigma-Aldrich, St.Louis, MO), unless otherwise noted.

**Generating PfPKG biosensors.** The sequence encoding Yellow PfPKG (PfPKG D domain S403-E542, flanked by CFP and Venus) and the CFP/Cp$^{173}$Venus-Venus biosensor variant (PfPKG D domain S403-E542 flanked by CFP and Cp$^{173}$Venus-Venus), synthesized by Genscript, were moved into pcDNA3.1(-). The Yellow PfPKG biosensor was then packaged in adenovirus type 5, amplified and titer measured by VectorBiolabs (Malvern, PA). To obtain the Red PfPKG biosensor, the PfPKG D-domain sequence (S403-E542) was synthetized flanked by SphI/SacI restriction sequences and inserted into SphI/SacI-opened pUC57 vector by Genscript. The two fluorophores were ligated by inserting XhoI/SphI-cut T-sapphireΔC11 (T-sapphire)[48] fragment N-terminally and EcoRI/SacI-cut Dimer2[49] fragment C-terminally into SphI/SacI-opened PfPKG. Dimer2 (evolved from DsRed) is thus transcribed as a monomer, but has been shown to associate into a dimer[49]. Both fluorophores were excised from a Red cGES DE5 biosensor plasmid (synthesized by Genscript). The correct sequence was then moved into the expression vector pcDNA3.1(-). The Red cGES DE5 biosensor was packaged in adenovirus type 5, amplified and titer measured by VectorBiolabs (Malvern, PA).

**Constructing the His-TEV-Yellow PfPKG biosensor.** A His-7-TEV N-terminal tag was amplified from pQTEV (Addgene #31291) using forward and reverse primers (GGAGACCCAAGCTGGCTAGCATGAAACATCACCATCACCATC and CACAGGTACCGCAAGCTCGAATCCCTGAAAATAAAGATTCTC, respectively) and inserted N-terminally into a NheI- and XhoI-opened Yellow PfPKG vector using In-Fusion HD Cloning Kit (Takara Bio, Mountain View, USA, Inc.) according to the manufacturer's protocol.

**Mutagenesis.** Point mutations in the PfPKG Red biosensor were performed using QuikChange II Site-Directed Mutagenesis Kit (Agilent technologies, Santa Clara, California, USA) according to the manufacturer's protocol.

**Transfection of HEK293 cells and in vitro FRET assay.** HEK293 cells (ATCC) were cultured in Dulbecco's modified Eagle's medium with 10% fetal bovine serum, penicillin (100 U/ml) and streptomycin (100 µg/ml) (ThermoFischer, Waltham, Massachusetts, USA). Cells were transiently transfected with the indicated plasmids using LipofectAMINE 2000 (ThermoFischer) according to the manufacturer's protocol. After transfection, cells were either cultured on poly-lysin-coated glass cover-slides for FRET and confocal imaging (below) or cultured for 48 h for the in vitro FRET assay, then homogenized with Ultra-Turrax (Franke & Kunkel KG, Staufen, Germany) for 30 s at 4 °C in Buffer A (HEPES 10 mM pH 7.3, NaCl 137 mM, KCl 5.4 mM, CaCl$_2$ 2 mM, MgCl$_2$ 1 mM). Homogenates were stimulated with increasing concentrations of cGMP or cAMP in the presence of 100 µM IBMX. The donor fluorophore CFP was excited at 430 nm (430/24 filter), T-Sapphire was excited at 405 nm, (405/8 filter) and the fluorescence emission (F) of Venus and CFP (535/25 nm and 470/24 nm, respectively) or Dimer2 and T-Sapphire (590/20 nm and 515/30 nm, respectively) were measured on an EnVision plate reader (PerkinElmer, Melville, NY, USA). To visualize a positive response to cGMP, the FRET response was calculated as changes in $F_{CFP}/F_{Venus}$ and $F_{T-Sapphire}/F_{Dimer2}$. Emission spectra of the biosensors in homogenates were measured on a Synergy H1 plate reader (BioTek Instruments, Winooski, VT, USA) through excitation of both biosensors at 405 nm and recording emission from 450 to 670 nm.

**Purification of Yellow PfPKG biosensor.** His-TEV-Yellow PfPKG biosensor was purified using a 5 ml HisTrap HP column supplied with Ni$^{2+}$ ions (GE Healthcare, UK) coupled to a workstation and a BioLogic recorder (BIO-RAD). Prior to use, the column was pre-equilibrated with phosphate buffered saline with 0.05% sodium azide (PBS/0.05% sodium azide). Homogenate of HEK293 cells expressing the His-TEV-Yellow PfPKG biosensor was centrifuged at 4000 rpm for 40 min and the supernatant filtered with a 0.22 µm filter. This was applied to the HisTrap column with a flow of 0.5 ml/min, followed by washing using 150 ml of PBS/0.05% sodium azide and 50 ml of 25 mM Imidazole/PBS. The bound biosensors were eluted with 50 ml 250 mM imidazole/PBS (pH 7.3). The proteins were up-concentrated and buffer changed to Buffer A using Amicron Ultra-30 columns (Millipore).

**Animals.** Animal care was according to the Norwegian Animal Welfare Act (and approved by the Norwegian Animal Research Authority) and Use of Laboratory Animals (publication no. 85-23, revised 1996) and Animals (Scientific Procedures) Act 1986 (United Kingdom) which both conform to the European Convention for the protection of Vertebrate animals used for Experimental and other Scientific Purposes (Council of Europe no. 123, Strasbourg 1985).

**Isolation of cardiomyocytes.** Primary culture and transduction of cardiac ventricular myocytes from male Wistar rats were prepared[27], attached to laminin-coated cover slides and transduced with adenovirus containing the biosensor as previously described[50,51].

**Isolation of stellate ganglion neurons.** Three- to four-week old male Wistar rats were sacrificed by an approved Home Office Schedule 1 method comprising (under isoflurane anesthesia) intraperitoneal pentobarbitone overdose (500 mg/kg) followed by exsanguination. Both stellate ganglia were excised intact and placed into Ca$^{2+}$- and Mg$^{2+}$-free Hanks' Balanced Salt Solution on ice, where any contaminating tissue was dissected away before they were each cut into 6–8 pieces. These next underwent enzymatic digestion in collagenase IV (22 min) and trypsin (25 min) at 37 °C, before two 5 min incubations in blocking buffer (87.12% Leibovitz's L-15 medium, 0.54% D-(+)-glucose solution, 1.8 mM L-glutamine, 90 units/ml penicillin, 90 µg/ml streptomycin and 10% fetal bovine serum). Finally the ganglionic tissue was mechanically triturated in neuronal plating medium (90% L-15 medium, 24 mM NaHCO$_3$, 38 mM glucose, 50 units/ml penicillin, 50 µg/ml streptomycin, 50 µM NGF and 10% foetal bovine serum) using a glass Pasteur pipette with a fire-polished tip to produce a single cell suspension, which was seeded onto 6 mm poly-D-lysine/laminin coated glass coverslips. These were kept at 37 °C under 5% CO$_2$ in neuronal plating medium.

**Confocal microscopy.** Twenty four to forty eight hours after HEK293 transfection or cardiomyocyte transduction, cells were incubated in Buffer B (NaCl 144 mM, KCl 1.97 mM, CaCl$_2$ 1 mM, MgCl$_2$ 1 mM, KH$_2$PO$_4$ 0.43 mM, K$_2$HPO$_4$ 1.5 M, glucose 10 mM) and visualized under an Olympus FV1000/BX61 confocal microscope using a water immersion objective (×60 1.1 NA). Cells were sequentially excited with a 458 nm laser for CFP and 515 nm laser for Venus, 488 nm laser for T-sapphire and 543 nm laser for Dimer2 and the emission was measured by Olympus Fluoview 1000 at 475/25 nm and 527/100 nm, for CFP and Venus, respectively, and at 510/30 nm and 581/100 nm for T-sapphire and Dimer2, respectively.

**FRET imaging.** Twenty four to forty eight hours after HEK293 transfection or cardiomyocyte transduction, cells were placed into a watertight cell imaging chamber (Attofluor, ThermoFischer) at room temperature in Buffer A (HEK293 cells) or Buffer B (cardiomyocytes). Cells expressing the biosensors were visualized through a motorized digital inverted fluorescent microscope (iMIC; FEI, Graefelfing, Germany) with an oil objective (×60 1.35 NA for HEK293 cells and ×40 1.35 NA for cardiomyocytes; Olympus, Tokyo, Japan). Fluorescent molecules were illuminated by a monochromator with fast-switching wavelengths (Polychrome V; FEI) for 20–80 ms at a frequency of 0.1–1 Hz. Cells expressing the red biosensors were excited for 80 ms at 422 ± 15 nm and 572 ± 15 nm consecutively and emission from T-sapphire and Dimer 2 were separated using a dichrotome iMIC Dual Emission Module, where a 560LP filter separates the two fluorophores' images on a single EM-CCD camera chip (EVOLVE 512; Photometrix, Tucson, USA). Fluorescence, from excitation at 422 nm, was measured at 537 ± 29 nm for T-sapphire and at 623 ± 25 nm for Dimer2 emission. FRET was measured as ratios of T-sapphire over Dimer2 (T-sapphire/Dimer2 ratio). Dimer2 emission was corrected for spillover of T-sapphire emission into the Dimer2-channel (15.1%). Cells expressing the yellow biosensor were excited at 436 ± 10 nm for 20–80 ms and emission was split using a dichrotome iMIC Dual Emission Module with a DCLP 505 filter that separates the two fluorophores onto a camera (EVOLVE 512). Fluorescence was measured at 530 ± 15 nm for Venus and at 470 ± 12 nm for CFP emission. Images were acquired by Live Acquisition browser (FEI) and FRET was calculated using Offline Analysis software (FEI). FRET was measured as ratio of CFP over Venus (CFP/Venus ratio). Venus emission was corrected for spillover of CFP emission into the Venus channel (63%). FRET was adjusted for photobleaching and representative experiments were run through a Savitzky–Golay filter for clarity using GraphPad Prism 7 for Windows (GraphPad Software, San Diego, CA).

Stellate neurons were transduced with adenovirus encoding the Yellow PfPKG or cGi-500[18] biosensors and imaged 2–3 days later at room temperature under ~2–3 ml/min gravity-driven perfusion with Tyrode's solution (NaCl 135 mM, KCl 4.5 mM, HEPES 20 mM, Glucose 11 mM, MgCl$_2$ 1 mM, CaCl$_2$ 2 mM) in a 100 µL chamber. An inverted fluorescent microscope (Nikon, Tokyo, Japan) with a 40x oil objective was used for such imaging. The cells were illuminated at 430 nm by an OptoLED light source (Cairn Research Ltd, Faversham, UK) and the emission was passed through DV2 beam-splitter (Photometrics) with the 05-EM filter set, consisting of a 505DCXR dichroic mirror and specific D480 ± 15 nm (CFP) & D535 ± 20 nm (Venus) emission filters (Chroma Technology Corp.). Images were captured every 15 s by a CoolSNAP HQ2 digital CCD camera (Photometrics) and

processed using OptoFluor CoolSNAP software (Cairn Research Ltd.) to record changes in CFP and Venus emission under 430 nm excitation during drug treatment protocols. Changes in the FRET ratio (CFP/Venus) were used as an index of changing cGMP concentration.

**Total cGMP measurements.** Isolated ventricular cardiomyocytes (cultured for 48 h to ensure similar conditions as cells used for FRET imaging) and HEK293 cells were stimulated in the presence and absence of 50 nM SNAP, 100 µM SNAP and/ or 300 nM CNP (as indicated) for 10 min. Cyclic GMP levels were measured using a cGMP enzyme immunoassay kit (Cayman Chemical Company, Ann Arbor, MI, USA) as previously described[36].

**NO release measurements.** Intracellular release of NO was measured by DAF-FM Diacetate (DA), a membrane permeable dye that reacts with NO to form a fluorescent benzotriazole. Cardiomyocytes were plated on laminin-coated cover slides and after 2 h loaded with 20 µM DAF-FM DA in the dark at 37 °C for 30 min. After washing, cover slides were placed into a watertight cell imaging chamber (Attofluor, Thermo Fischer) at room temperature in Buffer B. Cells were visualized through a motorized digital inverted fluorescent microscope (iMIC; FEI, Grae-felfing, Germany) with an oil immersion objective (×40 oil 1.35 NA; Olympus, Tokyo, Japan). Cells were excited at $500 \pm 10$ nm and emission was measured at $530 \pm 15$ nm.

**Statistics and reproducibility.** Statistical significance was determined by Two-tailed Student's $t$ test, Two-tailed Mann–Whitney test, One-way ANOVA with Dunnett's post hoc test, two-way ANOVA with Sidak's multiple comparisons test and Tukey's multiple comparisons test where indicated using GraphPad Prism 7 and 8 for Windows.

**Reporting summary.** Further information on research design is available in the Nature Research Reporting Summary linked to this article.

## Data availability
Nucleotide sequence for Yellow *Pf*PKG and Red *Pf*PKG have been deposited in GenBank under the accession numbers MK496217 and MK496218, respectively. The data that support the findings of this study are available from the corresponding author upon request. The authors declare that the data supporting the findings of this study are available within the paper and its supplementary information files.

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

## Acknowledgements

The authors thank Iwona Gutowska-Schiander for technical assistance. This work was supported by the South-Eastern Norway Regional Health Authority, the Norwegian Council on Cardiovascular Diseases, The Research Council of Norway, Stiftelsen Kristian Gerhard Jebsen, The Anders Jahre Foundation for the Promotion of Science, The Simon Fougner Hartmann Family Foundation, The Family Blix Foundation and grants from the University of Oslo. C.K. is funded by the NIH grant R01 GM090161.

## Author contributions

G.C., D.L., A.H.U., D.P., C.K., F.O.L. and K.W.A. designed research; G.C., D.L., A.H.U., J.J.K., O.C.N., L.R.M., M.B. performed research; G.C., D.L., A.H.U., J.J.K., O.C.N., L.R.M., M.B., D.P., C.K., F.O.L., K.W.A. analyzed data; G.C., F.O.L., K.W.A. wrote the paper; all authors revised and approved the manuscript.

## Competing interests

The authors have no competing interests to declare.
