## [Peer Review File · Communications Biology]

Reviewers' comments:

Reviewer #1 (Remarks to the Author):

This is a nice report describing the development of characterization of the new high affinity FRET biosensors for cGMP based on D domains of cGMP dependent proteins kinase from *Plasmodium falciparum*. This is a nice development for the field since it adds a new (or even two new) high affinity cGMP biosensors to our toolbox which had contained just one red cGES-DE5 that far, capable of measuring cGMP in low-middle nanomolar range. Here, the authors present a clear rationale and a straight forward "rational design" approach and thoroughly characterize new sensors in vitro and in live cells (HEK293 line and primary cardiomyocytes and SG neurons), two latter cell types contain endogenously low levels of cGMP. The paper was well written. Authors are lauded for the development of new sensor and for presenting clear data, and for their honestly dealing with issues of challenging cGMP measurements and sensor development for low nm range. Since it is a clear advance for the field I would recommend acceptance with minor (mostly text revisions):

1. Figure 2b, scale bar. Is it really 100 μm ? which would mean huge HEK cells. Please check if not mistaken, e.g. 10 μm
2. traces 3e-h are representative with no data analysis / bar graphs shown. consider showing all data points on a graph.
3. Fig 5f - datapoint represented on the trace (around 15% CNP response) is not shown in the graph fig 5g - pls explain the discrepancy/ fix
4. Many fig legend state to exact n numbers but a range.g. 8-16 cells, while it is clear that one group contained 8, another one 16 measurements - pls state exact numbers.
5. At several instances n numbers are too low. e.g. "3-4- cells from 2-3 animals" which is in my opinion not solid enough. pls add some more data points in such cases.

Reviewer #2 (Remarks to the Author):

In this paper, Calamera and colleagues describe the development of FRET-based cGMP biosensors which are suited to detect the changes in low nanomolar cGMP concentrations. The authors have accomplished this purpose by using the cGMP binding domain D of *Plasmodium falciparum* PKG which is known for high cGMP affinity, and monitored subtle cGMP productions in ganglion neurons and cardiomyocytes using one of the resulting sensors, Yellow PfPKG.

I have found two novel points in this paper: i) PfPKGs are the first sensors with the protozoan parasite's cGMP binding domain. Mammalian PKGs or PDEs have been used in the previously reported cGMP biosensors; ii) Yellow PfPKG is the first sensor based on CFP-YFP FRET which has the high cGMP affinity as well as Red cGES-DE5. Although its selectivity for cGMP/cAMP is obviously lower than of Red cGES-DE5, the dynamic range is larger. The new varieties of cGMP biosensors would be welcome in the research community for cell signaling.

(1) It is unclear how to calculate FRET ratio for T-Sapphire-Dimer2 sensors in live cell imaging. In Materials and Methods section, the authors describe "Cells expressing the red sensors were excited at 422 ± 15 nm and 572 ± 15 nm" (lines 197-198) but "Cells expressing the yellow sensor were excited at 436 ± 10 nm" (lines 205-206). Why were two excitation wavelengths needed for the red sensors? The Dimer2 emission excited at 572 nm shouldn't be used as the denominator of FRET ratio because this

emission doesn't depend on cGMP concentrations. If this emission was used for correction for "direct excitation at 422 nm (1%)" (lines 202-203), the yellow sensors should also be excited at ~515 nm for the correction for a fair comparison.

(2) Although there are many figures for time course of live cell FRET imaging in this paper, Figs. 4a-b are the only panels which have the vertical axis with "% of max". I think that this axis should be changed with the axis with "CFP/YFP Ratio (normalized)" as well as the other figures because the actual changes of FRET ratio are important for signal detection in this case.

Also, the result of a statistical test should be shown for quantitative comparison of Fig. 4a described as "As predicted, BNP stimulation increased FRET significantly more for the Yellow PfpPKG than for the cGi-500 biosensor" (lines 331-332).

(3) As the authors also noticed, 100-fold selectivity for cGMP/cAMP is not high in high affinity sensors. cAMP affinities of PfpPKGs (the Yellow sensor and the Red sensor with EC_{50} 4.6 μ M and 1.9 μ M, respectively) are comparable to of some cAMP biosensors. For example, EC_{50} values of Epac1-camps (Nikolaev et al., 2004) and Epac-S^{H134} (Klarenbeek et al., 2015) are 2.4 μ M and 4.0 μ M, respectively. It means that the cAMP sensors are helpful to check whether the signal changes of PfpPKGs are due to cAMP or not. I recommend including the discussion about the combined use with cAMP biosensors. Especially, Yellow PfpPKG would be suitable for coexpression with single-wavelength red fluorescent cAMP biosensors such as Pink Flamingo (Harada et al., 2017).

Minor comments.

1) Emission spectra with and without cGMP of PfpPKGs are informative for users. Adding it into Fig. 1 (or Supplementary Figure) might be helpful.

2) The FRET signal change of Red cGES-DE5 finally reached ~18% in Fig. 5f, but I can't find the plot in Fig. 5g (Iso+CNP+IBMX).

3) Although the manuscript says "However, the dynamic range of the cGi-500 was larger than the PfpPKG sensor (Figure 4c)." (lines 431-432), I think that basal cGMP levels in cells might reduce the maximal FRET responses of Yellow PfpPKG which has the high affinity.

4) Since the dynamic range of Red PfpPKG seems small, comparable to of Red cGES-DE5, I think that the representation of "a large dynamic range" in lines 455-456 ("To conclude, we have generated two novel biosensors for cGMP with high affinity and a large dynamic range.") is not appropriate.

Reviewer #3 (Remarks to the Author):

The manuscript by G. Calamera et al. presents two new genetically-encoded sensors (Yellow PfpPKG and Red PfpPKG) that are designed for measuring nanomolar concentrations of cGMP. Only one sensor was available so far that could provide similar sensitivity (Red cGES-DE5). Yellow PfpPKG shows higher dynamic range and ~2.5 higher affinity than Red cGES-DE5 in cardiomyocytes. Yellow PfpPKG also performs well in stellate ganglion neurons, when compared to another known sensor, cGi-500. These data demonstrate the significance of the study. The paper is well written, sound, and contains necessary control experiments. Together with positive experimental results, the authors also describe the experiments, where their sensors failed to surpass the performance of the known sensors. This information is important because it can save time for potential users.

A serious improvement of the paper can be expected if the authors would measure the cGMP concentration dependence in purified solutions. The cell homogenates that they use now contain

unknown basal concentration of cGMP and also non-negligible autofluorescence and scattering. These effects can introduce systematic and random errors in their measurements of the EC50 and dynamic range.

Several minor issues should also be addressed.

1. The authors should specify what protein corresponds to an abbreviation Dimer 2. What red FP is a monomer in this dimer? Is this tandem dimer or physically associated dimer without peptide link?
2. It would be easier for the reader to see all EC50 values presented in the same units, namely either nM or μ M (lines 88-90);
3. Please spell out EC50 and add it to the list of abbreviations;
4. Please add the abbreviation of stellate ganglion (SG) to the list as well;
5. It seems that the method of SG neurons transduction is missing;
6. When comparing the performance of Cp173Venus-Venus with Yellow PfPKG (lines 277-278), the authors say that they found no alteration of affinity to cGMP, although EC50 of Cp173Venus-Venus is ~ 2 times lower than that of Yellow PfPKG (46 vs 22 nM). Is this correct?
7. It also looks like Yellow PfPKG has ~ 2.5 times higher affinity compared to Red cGES-DE5 in intact cardiomyocytes (lines 322-323). That makes Yellow PfPKG even a better sensor. However, the authors statement is that they have equal sensitivity. Is the 2.5 x difference insignificant statistically? This should be clarified.
8. When comparing sensitivity of Yellow PfPKG to cAMP and cGMP it would be helpful to show the titration curves on the same plot (For example, combining data of Figs. 1c and 5d).
9. Concentration of forskolin should be given in line 347.
10. In Discussion, lines 381-383, different dynamic ranges of Yellow and Red sensors are attributed to different efficiency of transferring energy from donor to acceptor. It is rather different relative change of energy transfer upon binding cGMP that explains different dynamic ranges.
11. In lines 388-389, it is better to say that longer excitation wavelength, instead of emission wavelength, can penetrate deeper, etc. The bluer emission wavelength will experience more scattering events, but still can rich the detector with the high enough numerical aperture.
12. Lines 401-402: See comment 7.

Reviewer #1 (Remarks to the Author):

This is a nice report describing the development of characterization of the new high affinity FRET biosensors for cGMP based on D domains of cGMP dependent proteins kinase from Plasmodium falciparum. This is a nice development for the field since it adds a new (or even two new) high affinity cGMP biosensors to our toolbox which had contained just one red cGES-DE5 that far, capable of measuring cGMP in low-middle nanomolar range. Here, the authors present a clear rationale and a straight forward "rational design" approach and thoroughly characterize new sensors in vitro and in live cells (HEK293 line and primary cardiomyocytes and SG neurons), two latter cell type contain endogenously low levels of cGMP. The paper was well written. Authors are lauded for the development of new sensor and for presenting clear data, and for their honestly dealing with issues of challenging cGMP measurements and sensor development for low nm range. Since it is a clear advance for the field I would recommend acceptance with minor (mostly text revisions):

1. Figure 2b, scale bar. Is it really 100 μm ? which would mean huge HEK cells. Please check if not mistaken, e.g. 10 μm

Thanks for noticing the mistake. Now corrected to 10 μm

2. traces 3e-h are representative with no data analysis / bar graphs shown. consider showing all data points on a graph.

We agree that it is more informative for the reader to show the data points from all experiments and not only representative examples. We therefore added data from Figure 3b, e-g into Figure 3j. The data from CNP concentration response curves are collapsed and shown in Figure 3i.

3. Fig 5f - datapoint represented on the trace (around 15% CNP response) is not shown in the graph fig 5g - pls explain the discrepancy/ fix

Traditionally, FRET measurements have been calculated as $F(\text{Acceptor})/F(\text{Donor})$. Since this produces a decrease in FRET with increasing concentration of cGMP for all the sensors used in this study, we decided to plot all traces as $F(\text{Donor})/F(\text{Acceptor})$ for pedagogical purposes (increases in cGMP would translate into increased ratio), as performed by ^{1, 2} and others. Unfortunately, the bar graphs represented the percentage change in $F(\text{Acceptor})/F(\text{Donor})$. We have now reanalyzed all data as described in Materials and Methods and plotted this as $F(\text{Donor})/F(\text{Acceptor})$. This changed the magnitude of the responses but not the conclusion from each experiment.

4. Many fig legend state to exact n numbers but a range.g. 8-16 cells, while it is clear that one group contained 8, another one 16 measurements - pls state exact numbers.

Now the figure legends report clearly the number for each experiment type

5. At several instances n numbers are too low. e.g. "3-4 cells from 2-3 animals" which is in my opinion not solid enough. pls add some more data points in such cases.

We have now performed more experiments to increase the number of animals and number of cells.

Reviewer #2 (Remarks to the Author):

In this paper, Calamera and colleagues describe the development of FRET-based cGMP biosensors which are suited to detect the changes in low nanomolar cGMP concentrations. The authors have accomplished this purpose by using the cGMP binding domain D of *Plasmodium falciparum* PKG which is known for high cGMP affinity, and monitored subtle cGMP productions in ganglion neurons and cardiomyocytes using one of the resulting sensors, Yellow PfPKG.

I have found two novel points in this paper: i) PfPKGs are the first sensors with the protozoan parasite's cGMP binding domain. Mammalian PKGs or PDEs have been used in the previous reported cGMP biosensors; ii) Yellow PfPKG is the first sensor based on CFP-YFP FRET which has the high cGMP affinity as well as Red cGES-DE5. Although its selectivity for cGMP/cAMP is obviously lower than of Red cGES-DE5, the dynamic range is larger. The new varieties of cGMP biosensors would be welcome in the research community for cell signaling.

(1) It is unclear how to calculate FRET ratio for T-Sapphire-Dimer2 sensors in live cell imaging. In Materials and Methods section, the authors describe "Cells expressing the red sensors were excited at 422 ± 15 nm and 572 ± 15 nm" (lines 197-198) but "Cells expressing the yellow sensor were excited at 436 ± 10 nm" (lines 205-206). Why were two excitation wavelengths needed for the red sensors? The Dimer2 emission excited at 572 nm shouldn't be used as the denominator of FRET ratio because this emission doesn't depend on cGMP concentrations. If this emission was used for correction for "direct excitation at 422 nm (1%)" (lines 202-203), the yellow sensors should also be excited at ~ 515 nm for the correction for a fair comparison.

In the original manuscript, the correction of "direct excitation at 422 nm (1%)" is based on 422nm-excitation of Dimer2 being 1% of its 572nm excitation. Please note that excitation at 422nm and 572nm was performed sequentially but with the same illumination times to allow this correction. The same correction could not be performed with the Yellow PfPKG biosensor since direct excitation of Venus (515nm), at the illumination times chosen for the considerably weaker CFP, reached saturation, thus precluding meaningful correction of direct excitation of Venus.

The Donors (T-sapphire and CFP) and acceptors (Dimer2 and Venus) are in a 1:1 stoichiometry and the direct excitation of acceptor at donor excitation (422nm and 436nm) will be similar in each cell and throughout each experiment and therefore this correction is not necessary for comparing changes in FRET, similar to other labs using similar methods^{1,3}.

Since correction of direct excitation of acceptor (at donor excitation) was only performed with the Red biosensors, we therefore agree with the reviewer that a direct comparison of the maximal FRET obtained for the Red and Yellow PfPKG was not a fair comparison. To make a fair comparison, we have therefore analyzed the Red PfPKG and Red-cGES-DE5 responses as ratios of T-sapphire over Dimer2 with Dimer2 emission only corrected for spillover of T-sapphire into the Dimer2 channel (15%). This has now been corrected in lines 219-222 and Figure 2e, 3i-k and 5g. The reanalysis did not result in large changes in data or conclusions from the experiments.

(2) Although there are many figures for time course of live cell FRET imaging in this paper, Figs. 4a-b are the only panels which have the vertical axis with "% of max". I think that this axis should be changed with the axis with "CFP/YFP Ratio (normalized)" as well as the other figures because the actual changes of FRET ratio are important for signal detection in this case.

Also, the result of a statistical test should be shown for quantitative comparison of Fig. 4a described as "As predicted, BNP stimulation increased FRET significantly more for the Yellow PfPKG than for the cGi-500 biosensor" (lines 331-332).

We originally presented the traces as % of max because we thought that it would be a better comparison between the two sensors since they display a different dynamic range (cGi-500 having a larger dynamic range than the Yellow PfPKG; Figure 4h).

However, we agree with the reviewer that consistency between figures is better for the reader and we have therefore modified the figure to show CFP/Venus ratio (normalized); for Figure 4a, d or CFP/YFP ratio (normalized); for Figure 4b, e.

We have also added the quantification of the individual responses and we have added the data points into bar graphs. The statistical difference between PfPKG and cGi-500 subjected to BNP-stimulation are now performed and indicated in Figure 4c and changes are made in the text (lines 357-9).

(3) As the authors also noticed, 100-fold selectivity for cGMP/cAMP is not high in high affinity sensors. cAMP affinities of PfPKGs (the Yellow sensor and the Red sensor with EC₅₀ 4.6 μM and 1.9 μM, respectively) are comparable to of some cAMP biosensors. For example, EC₅₀ values of Epac1-camps (Nikolaev et al., 2004) and Epac-SH134 (Klarenbeek et al., 2015) are 2.4 μM and 4.0 μM, respectively. It means that the cAMP sensors are helpful to check whether the signal changes of PfPKGs are due to cAMP or not. I recommend including the discussion about the combined use with cAMP biosensors. Especially, Yellow PfPKG would be suitable for coexpression with single-wavelength red fluorescent cAMP biosensors such as Pink Flamingo (Harada et al., 2017).

Thank you for the suggestion. We agree that a single-wavelength cAMP biosensor would be the best option for this purpose. We have now included this possible use of the sensor in the discussion (lines 482-485).

Minor comments.

1) Emission spectra with and without cGMP of PfPKGs are informative for users. Adding it into Fig. 1 (or Supplementary Figure) might be helpful.

Thank you for the suggestion. We have measured the emission spectra for both PfPKG sensors with and without cGMP and added into Figure 1 as suggested. We agree that this adds useful information about the sensors.

2) The FRET signal change of Red cGES-DE5 finally reached ~18% in Fig. 5f, but I can't find the plot in Fig. 5g (Iso+CNP+IBMX).

Traditionally, FRET measurements have been calculated as $F(\text{Acceptor})/F(\text{Donor})$. Since this produces a decrease in FRET with increasing concentration of cGMP for all the sensors used in this study, we decided to plot all traces as $F(\text{Donor})/F(\text{Acceptor})$ for pedagogical purposes (increases in cGMP would translate into increased ratio), as performed by ^{1, 2} and others. Unfortunately, the bar graphs represented the percentage change in $F(\text{Acceptor})/F(\text{Donor})$. We have now reanalyzed all data as described in Materials and Methods and plotted this as $F(\text{Donor})/F(\text{Acceptor})$. This changed the magnitude of the responses but not the conclusion from each experiment.

3) Although the manuscript says "However, the dynamic range of the cGi-500 was larger than the PfPKG sensor (Figure 4c)." (lines 431-432), I think that basal cGMP levels in cells might reduce the maximal FRET responses of Yellow PfPKG which has the high affinity.

We also considered if basal cGMP levels within the measurable range of this biosensor would limit the maximal FRET responses. However, when we applied ODQ to inhibit basal sGC activity in

cardiomyocytes (figure 3c) there was not a decrease in FRET, suggesting that cGMP levels, at least in cardiomyocytes, were below the range of detection with this biosensor.

4) Since the dynamic range of Red PfPKG seems small, comparable to of Red cGES-DE5, I think that the representation of “a large dynamic range” in lines 455-456 (“To conclude, we have generated two novel biosensors for cGMP with high affinity and a large dynamic range.”) is not appropriate.

We agree and thus modified the text (lines 486-7).

Reviewer #3 (Remarks to the Author):

The manuscript by G. Calamera et al. presents two new genetically-encoded sensors (Yellow PfPKG and Red PfPKG) that are designed for measuring nanomolar concentrations of cGMP. Only one sensor was available so far that could provide similar sensitivity (Red cGES-DE5). Yellow PfPKG shows higher dynamic range and ~2.5 higher affinity than Red cGES-DE5 in cardiomyocytes. Yellow PfPKG also performs well in stellate ganglion neurons, when compared to another known sensor, cGi-500. These data demonstrate the significance of the study. The paper is well written, sound, and contains necessary control experiments. Together with positive experimental results, the authors also describe the experiments, where their sensors failed to surpass the performance of the known sensors. This information is important because it can save time for potential users.

A serious improvement of the paper can be expected if the authors would measure the cGMP concentration dependence in purified solutions. The cell homogenates that they use now contain unknown basal concentration of cGMP and also non-negligible autofluorescence and scattering. These effects can introduce systematic and random errors in their measurements of the EC50 and dynamic range.

We thank the reviewer for the good evaluation and suggestion to improve the paper. We have now performed purification of the sensor using a protein purification protocol based on His-tag trapping column. We tested the purified sensor simultaneously with the cell homogenate expressing the His-tagged sensor for comparison and used the same protocol as previously. The affinity for cGMP (EC_{50}) was the same for the sensor in pure solution as in homogenates and comparable to that of the Yellow PfPKG sensor without the His-tag. The dynamic range of the His-tagged purified sensor and the Yellow PfPKG sensor in homogenates was not different ($42.3 \pm 0.4\%$ vs. $39.4 \pm 3.5\%$). We present these results as supplementary data but describe them in lines 279-284. Based on these data, we think that the potential influence of endogenous cGMP from HEK293 cells on the determined cGMP affinity of our biosensor is negligible.

Several minor issues should also be addressed.

1. The authors should specify what protein corresponds to an abbreviation Dimer 2. What red FP is a monomer in this dimer? Is this tandem dimer or physically associated dimer without peptide link?

A monomeric Dimer2 was taken from the established Red-cGES-DE5 biosensor⁴. Dimer2 is evolved from the tetrameric DsRed (through 17 mutations) and each monomer of Dimer2 can potentially form dimers⁵. To avoid confusion in the specification of the fluorescent protein we used in the red biosensors, we have now specified what corresponds to Dimer2 in the Material and Methods section (lines 142-3).

2. It would be easier for the reader to see all EC50 values presented in the same units, namely either nM or μ M (lines 88-90);

We have changed them all to μ M (lines 96-98)

3. Please spell out EC50 and add it to the list of abbreviations;

It is now added to the list of abbreviations

4. Please add the abbreviation of stellate ganglion (SG) to the list as well;

This has now been done

5. It seems that the method of SG neurons transduction is missing;

We have now specified that it was by adenoviral transduction (line 233-4)

6. When comparing the performance of Cp173Venus-Venus with Yellow PfPKG (lines 277-278), the authors say that they found no alteration of affinity to cGMP, although EC50 of Cp173Venus-Venus is ~ 2 times lower than that of Yellow PfPKG (46 vs 22 nM). Is this correct?

This is correct. However, the difference did not reach significance. We have inserted the P-value ($p=0.13$) into line 300+Table1. We have also modified the text (line 297-300) to better reflect this possible difference.

7. It also looks like Yellow PfPKG has ~2.5 times higher affinity compared to Red cGES-DE5 in intact cardiomyocytes (lines 322-323). That makes Yellow PfPKG even a better sensor. However, the authors statement is that they have equal sensitivity. Is the 2.5 x difference insignificant statistically? This should be clarified.

In the Yellow PfPKG, CNP has ~1.9-fold higher EC50 than the red cGES-DE5, but the difference did not reach statistical significance. We have now performed additional experiments with concentration-response curves for CNP to increase the statistical power. However, still no significant difference was found. The p-value ($p=0.37$) is now included in the figure legend. We have also modified the text (line 348) to better reflect this.

8. When comparing sensitivity of Yellow PfPKG to cAMP and cGMP it would be helpful to show the titration curves on the same plot (For example, combining data of Figs. 1c and 5d).

Thank you. We agree it is helpful for the reader and we have now modified it so that data points from Figure 1c and e are included in Figure 5A and b for comparison. This is also stated in the figure legend (line 728-9)

9. Concentration of forskolin should be given in line 347.

It is now specified (line 374).

10. In Discussion, lines 381-383, different dynamic ranges of Yellow and Red sensors are attributed to different efficiency of transferring energy from donor to acceptor. It is rather different relative change of energy transfer upon binding cGMP that explains different dynamic ranges.

Thanks for specifying this. We have now re-written the sentence (line 408-410).

11. In lines 388-389, it is better to say that longer excitation wavelength, instead of emission wavelength, can penetrate deeper, etc. The bluer emission wavelength will experience more scattering events, but still can rich the detector with the high enough numerical aperture.

Thank you for bringing this up.

The red PfPKG is excited at lower wavelengths (or similar range in terms of deep tissue imaging) than the Yellow PfPKG sensor (422nm vs. 436nm). Therefore, both sensors will experience similar scattering from the excitation. Based on the reference 40 and 41,^{6, 7} fluorescent proteins with emission in the range of 650-800nm are more suitable for deep-tissue imaging due to less

background absorbance from for example hemoglobin and myoglobin. Therefore, the Dimer2-emission would be better than Venus-emission in these applications. However, we agree with the reviewer that the sentence in lines 388-389 can be misunderstood, and we therefore rewrote it and also replaced the references to underscore our current argument (line 415-6).

12. Lines 401-402: See comment 7.

See our answer to comment 7

References

1. Subramanian, H. et al. Distinct submembrane localisation compartmentalises cardiac NPR1 and NPR2 signalling to cGMP. *Nat. Commun.* **9**, 2446 (2018).
2. Russwurm, M. et al. Design of fluorescence resonance energy transfer (FRET)-based cGMP indicators: a systematic approach. *Biochem. J.* **407**, 69-77 (2007).
3. Sprenger, J.U. et al. In vivo model with targeted cAMP biosensor reveals changes in receptor-microdomain communication in cardiac disease. *Nat Commun* **6**, 6965 (2015).
4. Niino, Y., Hotta, K. & Oka, K. Simultaneous live cell imaging using dual FRET sensors with a single excitation light. *PLoS One* **4**, e6036 (2009).
5. Campbell, R.E. et al. A monomeric red fluorescent protein. *Proc. Natl. Acad. Sci. U. S. A.* **99**, 7877-7882 (2002).
6. Chu, J. et al. Non-invasive intravital imaging of cellular differentiation with a bright red-excitable fluorescent protein. *Nat. Methods* **11**, 572 (2014).
7. Filonov, G.S. et al. Bright and stable near-infrared fluorescent protein for in vivo imaging. *Nat. Biotechnol.* **29**, 757 (2011).

REVIEWERS' COMMENTS:

Reviewer #1 (Remarks to the Author):

The authors have addressed all my comments satisfactorily, their ms is now acceptable for publication

Reviewer #2 (Remarks to the Author):

Most of my concerns were addressed in detail by the author's response, but I have a concern about newly-added panels Fig. 4c and 4f that respond to my comment (2).

I think that the vertical axis with "% of max" for these bar graphs should be changed with the axis with "% change".

These panels for the comparison between Yellow PfPKG and cGi-500 are corresponding to Fig. 3j and 5g for the comparison between Yellow PfPKG and Red cGES-DE5 (as well as Fig. 4g is corresponding to Fig. 3k).

"% of max" in these panels is not consistent with "% change" in Fig. 3j and 5g (I think "% of max" for Fig. 3i, 5a and 5b is proper), and the amplitude of the actual changes of FRET ratio is important for detecting subtle concentration changes and should be compared.

Reviewer #3 (Remarks to the Author):

The authors have considered all my issues and replied my questions. The paper can be accepted now.

Reviewer #1 (Remarks to the Author):

The authors have addressed all my comments satisfactorily, their ms is now acceptable for publication

Thank you for a constructive review process.

Reviewer #2 (Remarks to the Author):

Most of my concerns were addressed in detail by the author's response, but I have a concern about newly-added panels Fig. 4c and 4f that respond to my comment (2).

I think that the vertical axis with "% of max" for these bar graphs should be changed with the axis with "% change".

These panels for the comparison between Yellow PfPKG and cGi-500 are corresponding to Fig. 3j and 5g for the comparison between Yellow PfPKG and Red cGES-DE5 (as well as Fig. 4g is corresponding to Fig. 3k).

"% of max" in these panels is not consistent with "% change" in Fig. 3j and 5g (I think "% of max" for Fig. 3i, 5a and 5b is proper), and the amplitude of the actual changes of FRET ratio is important for detecting subtle concentration changes and should be compared.

First of all, thank you for a constructive review process.

We also thank you for pointing out that the newly added 4c and 4f are not consistent with the other figures. We have therefore revised according to your suggestions and replaced figures 4c and 4f with figures showing %change. We also changed lines 222-3 (in results) and 334-5 (in discussion) accordingly.

To enable the reader to evaluate the difference between the two biosensors when one considers that they have a different dynamic range (Figure 4g), we have moved the original Figure 4c and f into Supplementary Figure 3. The lines 337-9 (in discussion) were changed accordingly.

Reviewer #3 (Remarks to the Author):

The authors have considered all my issues and replied my questions. The paper can be accepted now.

Thank you for a constructive review process.